# Incomplete Polymerization of Dual-Cured Resin Cement Due to Attenuated Light through Zirconia Induces Inflammatory Responses

**DOI:** 10.3390/ijms24129861

**Published:** 2023-06-07

**Authors:** Takeru Kondo, Hiroaki Kakinuma, Kanna Fujimura, Sara Ambo, Koki Otake, Yumi Sato, Hiroshi Egusa

**Affiliations:** 1Division of Molecular & Regenerative Prosthodontics, Tohoku University Graduate School of Dentistry, Sendai 980-8575, Japan; 2Department of Next-Generation Dental Material Engineering, Tohoku University Graduate School of Dentistry, Sendai 980-8575, Japan

**Keywords:** dual-cured resin cement, zirconia, resin monomer, innate immune system, mitogen-activated protein kinase

## Abstract

Zirconia restorations are becoming increasingly common. However, zirconia reduces the polymerization of dual-cured resin cement owing to light attenuation, resulting in residual resin monomers. This study investigated the effects of dual-cured resin cement, with incomplete polymerization owing to attenuated light through zirconia, on the inflammatory response in vitro. The dual-cured resin cement (SA Luting Multi, Kuraray) was light-irradiated through zirconia with three thickness diameters (1.0, 1.5, and 2.0 mm). The light transmittance and the degree of conversion (DC) of the resin cement significantly decreased with increasing zirconia thickness. The dual-cured resin cement in 1.5 mm and 2.0 mm zirconia and no-irradiation groups showed significantly higher amounts of hydroxyethylmethacrylate and triethyleneglycol dimethacrylate elution and upregulated gene expression of proinflammatory cytokines *IL-1β* and *IL-6* from human gingival fibroblasts (hGFs) and *TNFα* from human monocytic cells, compared with that of the 0 mm group. Dual-cured resin cement with lower DC enhanced intracellular reactive oxygen species (ROS) levels and activated mitogen-activated protein (MAP) kinases in hGFs and monocytic cells. This study suggests that dual-cured resin cement with incomplete polymerization induces inflammatory responses in hGFs and monocytic cells by intracellular ROS generation and MAP kinase activation.

## 1. Introduction

Zirconia has been frequently used for prosthodontic treatment because of its high toughness and success rate [1]. In addition, the opaque effects of zirconia cores based on yttria-stabilized tetragonal zirconia polycrystals (Y-TZP) have been attributed to the masking of metal cores and dichromatic abutment teeth owing to their low absorption coefficients, high refractive indices, and high opacity [2]. Resin cement can be used to achieve a high bond strength of zirconia to abutment teeth or core materials, and a reliable cementation method using dual-cured resin cement has been established [3]. However, the high opacity of zirconia can reduce the degree of conversion (DC) of dual-cured resin cement owing to light attenuation. The transmitted light intensity significantly decreases as the thickness of zirconia increases, resulting in insufficient polymerization [4].

DC determines the mechanical and physical properties of dual-cured resin cement [5]. Dual-cured resin cement with inadequate polymerization may cause increased water sorption, lower color stability, faster marginal degradation, and loss of restoration [6,7]. The incomplete polymerization of resin cement also increases residual dimethacrylate monomers, such as triethyleneglycol dimethacrylate (TEGDMA) and hydroxyethylmethacrylate (HEMA), which are frequently used for dual-cured resin cement to reduce monomer viscosity and enhance the bonding strength of cement to dentin [8,9]. HEMA and TEGDMA are water-soluble [10], and residual monomers can be leached from the restored oral environment through saliva or gingival crevicular fluid [11]. Negative biocompatibility, particularly the cytotoxicity of TEGDMA and HEMA, has been intensively studied [12]. TEGDMA and HEMA induce cellular stress, DNA damage, and cell apoptosis by producing intracellular reactive oxygen species (ROS) [13,14]. Therefore, residual monomers from resin cement with inadequate polymerization may negatively affect periodontal tissue cells after the cementation of the prostheses.

The non-enzymatic antioxidant glutathione (GSH) depletion and the increase in intracellular ROS by resin monomers induce redox imbalance and activate mitogen-activated protein (MAP) kinases, which are closely related to cytokine release [15,16]. The biological effects of resin monomers on the innate immune system have been studied using human dental pulp stem cells (hDPSCs) because resin-based composites are clinically used directly with the dental pulp. Exposure to HEMA with ethanol for one day enhanced the production of IL-6 and TNFα in hDPSCs [17]. A one-day culture of hDPSCs with HEMA promoted the secretion of IFN-γ and IL-8, with higher intracellular ROS production and MAP kinase activation [18]. A three-day exposure to TEGDMA promoted the release of IL-8 from hDPSCs [19].

Based on this background, we hypothesized that dual-cured resin cement with residual resin monomers might induce inflammatory reactions in periodontal tissue cells. This study aimed to investigate the effects of zirconia on the physicochemical properties and residual resin monomers of dual-cured resin cement and the biological effects of dual-cured resin cement irradiated through zirconia on the innate immunology of periodontal tissue in vitro. This study provides experimental evidence that dual-cured resin cement with incomplete polymerization directly induces proinflammatory cytokine release in hGFs and human monocytic cells.

## 2. Results

### 2.1. Effects of Zirconia Thickness on the DC of the Dual-Cured Resin Cement through Light Attenuation

The total light transmittance decreased significantly with increased zirconia thickness (Figure 1a). The dual-cured resin cement that was irradiated through 1.5 mm and 2.0 mm thicknesses of zirconia and the resin cement with no irradiation showed significantly lower DC than the resin cement that was irradiated without zirconia (0.0 mm zirconia group) on day 1. On the other hand, the DC of the resin cement that was irradiated through a 1.0 mm thickness of zirconia was not significantly different compared to that of the 0.0 mm zirconia group. After day 2, the resin cement in the 1.5 mm and 2.0 mm zirconia and no-irradiation groups showed lower DC than resin cement in the 0.0 mm zirconia group, although the difference between the 1.5 mm and 0.0 mm zirconia groups was not significant (Figure 1b). These results indicate that >1.5 mm thickness of zirconia inhibits the polymerization of dual-cured resin cement through light attenuation.

### 2.2. Effects of Zirconia Thickness on Mechanical and Physical Properties of the Dual-Cured Resin Cement through Light Attenuation

The Vickers hardness of the dual-cured resin cement in the 1.5 mm and 2.0 mm zirconia and no-irradiation groups was significantly lower than that of the 0.0 mm zirconia group at all time points (Figure 2a). These results are consistent with the results of DC and indicate that dual-cured resin cement with lower DC exhibited lower surface stiffness. The surface roughness average of dual-cured resin cement in the 1.5 mm and 2.0 mm zirconia and no-irradiation groups was significantly higher than that in the 0.0 mm zirconia group, and these surface roughness averages gradually increased from day 1 to day 7 (Figure 2b). In addition, microcavities, which appear to be the traces of fillers, were observed on the surface of the resin cement in the 1.5 mm and 2.0 mm zirconia and no-irradiation groups; the number of microcavities in these groups increased from day 1 to day 7 (Figure 2c), which means that increasing the number of microcavities reflected the higher surface roughness averages. These results indicate that the dual-cured resin cement with lower DC caused microcavities, including the release of fillers in the water, and increased surface roughness.

### 2.3. Effects of Zirconia Thickness on Residual Resin Monomer Release of the Dual-Cured Resin Cement through Light Attenuation

HEMA and TEGDMA were detected in the supernatant of the dual-cured resin cement immersed in water. In contrast, bisphenol A-glycidyl dimethacrylate (Bis-GMA) and its core component, bisphenol A [20], were not detected in the supernatant (Figure 3a). The concentrations of released residual HEMA and TEGDMA were significantly higher in the 1.5 mm and 2.0 mm zirconia and no-irradiation groups than in the 0.0 mm zirconia group (Figure 3b,c). Moreover, the concentrations of HEMA and TEGDMA in the 1.5 mm and 2.0 mm zirconia and no-irradiation groups gradually increased from day 1 to day 7 (Figure 3b,c). These results suggest that the dual-cured resin cement with lower DC contained and released higher amounts of residual HEMA and TEGDMA in the water.

### 2.4. Effects of the Dual-Cured Resin Cement with Incomplete Polymerization on Human Gingival Fibroblasts (hGFs)

Human gingival fibroblasts (hGFs) were directly cultured on the dual-cured resin cement in each group. On day 2, the expression of proinflammatory cytokines, including *IL-1β*, *IL-6*, *PTGES2*, and *TNFα,* was significantly upregulated in the 2.0 mm zirconia and no-irradiation groups compared to the 0.0 mm zirconia group (Figure 4a). In addition, *MMP2* and *MMP9*, the matrix metalloproteinases associated with the extracellular matrix degeneration of gingival tissue [21], were upregulated in these groups, and the dual-cured resin cement in the 1.5 mm zirconia group also showed significantly higher expression of *MMP2* than the 0.0 mm zirconia group (Figure 4b). These results indicate that the dual-cured resin cement with lower DC induced the expression of proinflammatory cytokines and matrix metalloproteinases in hGFs. Furthermore, the cell number of hGFs in the 1.5 mm and 2.0 mm zirconia and no-irradiation groups was lower than that in the 0.0 mm zirconia group; the cell number in the 2.0 mm zirconia and no-irradiation groups decreased from day 1 to day 3 (Figure 4c). These results indicate the cytotoxicity of the dual-cured resin cement with incomplete polymerization of hGFs.

### 2.5. Effects of the Dual-Cured Resin Cement with Incomplete Polymerization on Human Monocytic Cells

THP-1 cells, the human monocytic cell line, were directly cultured on dual-cured resin cement in each group. On day 2, expression of the proinflammatory cytokine *TNFα* was significantly upregulated in the 1.5 mm and 2.0 mm zirconia and no-irradiation groups compared to the 0.0 mm zirconia group. In addition, the expression of *NOS2* in the no-irradiation group was significantly higher than that in the 0.0 mm zirconia group (Figure 5a). In contrast, the expression of anti-inflammatory cytokines, including *IL-10* and *ARG1,* was significantly downregulated in the 1.5 mm and 2.0 mm zirconia and no-irradiation groups compared to the 0.0 mm zirconia group (Figure 5b). These results indicate that dual-cured resin cement with lower DC polarized monocytic cells into a proinflammatory state. Furthermore, the number of THP-1 cells in the 1.5 mm and 2.0 mm zirconia and no-irradiation groups was significantly lower than that in the 0.0 mm zirconia group, and the cell number in these groups decreased from day 1 to day 3 (Figure 5c). These results suggest that dual-cured resin cement with incomplete polymerization showed higher cytotoxicity to human monocytic cells. However, the resin cement in the 0.0 mm and 1.0 mm zirconia groups showed a significantly lower cell number than the control polystyrene (Figure 5c), indicating that dual-cured resin cement is strongly cytotoxic to human monocytic cells, even when irradiated.

### 2.6. Effects of the Dual-Cured Resin Cement with Incomplete Polymerization on Intracellular ROS Generation and MAP Kinase Activation of hGFs and Human Monocytic Cells

The dual-cured resin cement in the 1.5 mm and 2.0 mm zirconia and no-irradiation groups induced significantly higher levels of intracellular ROS in hGFs and THP-1 cells 1 h after seeding the cells (Figure 6a,b). These results indicate that the dual-cured resin cement with lower DC induced higher levels of intracellular ROS in the hGFs and the THP-1 cells. In addition, the resin cement in the 0.0 mm and 1.0 mm zirconia groups showed significantly higher levels of intracellular ROS in hGFs and THP-1 cells compared to the control polystyrene (Figure 6a,b), suggesting that dual-cured resin cement, even when irradiated, could induce the generation of intracellular ROS.

To reveal the mechanism of the induction of proinflammatory cytokine expression in each cell type by dual-cured resin cement, we evaluated MAP kinase activation using the resin cement in the 2.0 mm zirconia group. ERK1/2 was phosphorylated at 1 h, and p38 was phosphorylated at 24 h in hGFs. JNK was phosphorylated at 1 h and peaked on day 1 (Figure 6c). The phosphorylation of ERK1/2 peaked on day 1, and p38 and JNK were immediately phosphorylated at 1 h in THP-1 cells (Figure 6d).

## 3. Discussion

The use of adhesive cementation with dual-cured resin cement significantly strengthens zirconia ceramic restorations, thereby impacting clinical performance. However, the biological effects of marginal cement after the cementation of the prostheses cannot be ignored. The cement margin width of a milled zirconia crown typically ranges from 42 to 151 µm [22,23,24]. However, a significant marginal degradation of resin cement after the cementation of prostheses was observed [25,26,27]. Indeed, the eluted resin monomers from the marginal cement of the crown restorations luted onto the tooth samples were detected even with enough irradiation [28]. Therefore, it seems that residual monomers were leached from marginal resin cement into periodontal tissue through saliva and crevicular fluid. Moreover, removing excessive resin cement around the prosthesis is extremely difficult because of its strong adhesion, leading to clinical complications. Cement remnants were found around the margins of all zirconia crowns despite the cleaning procedures [29]. These results led us to believe that the dual-cured resin cement with incomplete polymerization under zirconia restoration may remain in the gingival sulcus, especially in the deeper area. In this study, we cultured hGFs and THP-1 cells not on the supernatant of resin cement immersed in the medium but directly on the resin cement that imitates clinical situations. We demonstrated that the dual-cured resin cement that was irradiated through more than 1.5 mm of zirconia induced proinflammatory responses in the cells in human periodontal tissue (Figure 3 and Figure 4).

The thickness of the Y-TZP zirconia crown differs depending on the site of the crown. In particular, the thickness of the zirconia around the margin of the crown tends to be lower (0.8–1.0 mm) compared to occlusal reduction (1.5–2.0 mm) [30,31]. However, a fracture was sometimes observed around the margin of the zirconia crown because the zirconia’s thickness tended to be lower [32]. Large cracks were found in the 0.5 mm and 0.8 mm thicknesses of Y-TZP zirconia after fatigue but not in zirconia with a thickness greater than 1 mm [33]. In addition, an acceptable clinical color-masking ability of zirconia was achieved with a minimum thickness of 1.0 mm [34]. In this study, although the dual-cured resin cement that was irradiated through more than 1.5 mm of zirconia induced an inflammatory reaction, the 1.0 mm zirconia group exhibited DC, residual monomer release, and effects on inflammation with no significant difference from the 0.0 mm zirconia group (Figure 1, Figure 3, Figure 4 and Figure 5). One of our concerns is the release of resin monomers from marginal cement. Therefore, the results of previous studies and those of the present study suggest that a 1.0 mm thickness of preparation around the margin for the Y-TZP zirconia crown may be ideal for clinical applications.

ROS, generated through various extracellular stimuli, triggers proinflammatory cytokine production via signal transaction cascades, such as MAP kinases [35]. Resin-based materials induce intracellular ROS generation by residual resin monomers through the depletion of GSH, the principal intracellular antioxidant molecule [16,36]. The present study demonstrated that dual-cured resin cement with higher amounts of HEMA and TEGDMA release induced higher levels of intracellular ROS generation in hGFs and THP-1 cells one hour after seeding the cells (Figure 6). A study reported that the early elution of residual monomers, such as TEGDMA and UDMA, from dual-cured resin cement was detected one hour after the resin cement was immersed in water [9]. Therefore, HEMA and TEGDMA might have been eluted immediately after the resin cement was immersed in the culture medium in the present study and induced intracellular ROS generation. Bis-GMA can strongly induce intracellular ROS generation and cytotoxicity [37]. Although the dual-cured resin cement used in this study contained Bis-GMA (Table 1), Bis-GMA elution was not detected (Figure 3). Previous studies have also shown that Bis-GMA release was not detected in any dual-cured resin cement [38,39]. This can be explained by the low solubility of Bis-GMA in water [10]. Therefore, residual Bis-GMA may exist on the surface of the resin cement, as well as HEMA and TEGDMA, and these residual resin monomers may also induce intracellular ROS accumulation.

MAP kinases, including ERK1/2, p38, and JNK activated by toxic agents, regulate cellular responses, such as cell proliferation, apoptosis, and inflammatory cytokine production [40,41]. TEGDMA activated ERK1/2 after 30 min of exposure and JNK after 24 h of exposure in human pulp cells, and p38 was not activated by TEGDMA alone. On the other hand, in RAW264.7 cells, TEGDMA induced the phosphorylation of p38 and ERK1/2 after 24 h of exposure, whereas short-term exposure (15 min to 2 h) did not show activation, and JNK activation was not clearly detected in any time course [42]. Thus, each MAP kinase was differentially phosphorylated, depending on the exposure period and cell type [42]. The effects of resin cement on the MAP pathway in hGFs and THP-1 cells are poorly understood. In this study, we investigated the activation of MAP kinases in hGFs and THP-1 cells using dual-cured resin cement in the 2.0 mm zirconia group, which showed high amounts of residual monomers and the induction of intracellular ROS generation and proinflammatory cytokine expression in these cells. Most MAP kinases were activated in hGFs and THP-1 cells one hour after seeding the cells (Figure 6), suggesting that dual-cured resin cement with residual monomers may activate MAP kinases through intracellular ROS generation, resulting in proinflammatory cytokine production. Interestingly, the timing of phosphorylation differed depending on the MAP kinase and cell species (Figure 6), as reported in previous studies. These findings may shed light on the mechanisms by which resin materials induce the inflammatory reaction of hGFs and THP-1 cells. MAP kinases are reported to be upstream of the NF-κB signaling pathway, which plays an important role in the inflammatory response [43]. In addition, studies that investigated the relationship between ROS, signaling pathways, and cell function used antioxidants [16,44,45,46]. Therefore, further studies using antioxidants or investigating downstream MAP kinases are needed to elucidate the activation of signaling pathways in hGFs or THP-1 cells by resin cement through ROS generation.

Apart from residual monomers, the surface topography and stiffness of the resin cement might have affected inflammatory cytokine expression in this study because the cells were cultured directly on the resin cement. Physical environmental stimuli are sensed at the cell membrane and influence cell function and fate through mechanotransduction, which converts extracellular physical cues into intracellular biochemical signals transmitted to the cell [47,48,49,50,51]. When hGFs were cultured with polished or unpolished resins, the release of IL-6 and IL-8 was higher in the unpolished resin group than in the polished resin group [52]. The roughened titanium surface promoted the production of proinflammatory cytokines, such as TNF-α and Il-6, in RAW264.7 monocytic cells compared to the polished titanium surfaces [53]. Moreover, less-rough cobalt-chrome-molybdenum alloy discs suppressed the production of proinflammatory cytokines, including TNF-α and Il-6 in J774A.1 monocytic cells, compared to the more roughened discs [54]. Our study also showed that both hGFs and THP-1 cells, which were cultured on roughened resin cement, showed significantly higher expression of proinflammatory cytokines (Figure 2 and Figure 5). The stiffness of the local environment can tune cellular inflammatory responses through mechanotransduction [55]. When the hGFs were cultured on substrates with different stiffnesses (soft (5 kPa) or hard (25 kPa)), the expression of *PTGES2* and *IL-1β* was significantly enhanced in the soft substrate group compared to that in the hard substrate group [56]. Cultured mouse monocytes with different substrate stiffnesses showed higher expression of *Nos2* as stiffness increased [57]. Consistent with these results, our results demonstrated that the cultured cells on resin cement with lower Vickers hardness highly expressed proinflammatory cytokines (Figure 2 and Figure 5).

The design of this in vitro study possesses several limitations. One limitation arises from the correlation between in vitro experiments and clinical conditions. In clinical situations, the marginal thickness of the crown may be less than 1 mm [31]. To better stimulate these clinical situations, it would be beneficial to incorporate zirconia discs with thicknesses of 0, 0.8, 1.0, and 1.2 mm in this study. In vitro studies often employ a wide range of thicknesses to facilitate the clear detection of differences [58,59,60]. Therefore, we opted to select zirconia thicknesses of 0, 1.0, 1.5, and 2.0 mm in this study. Nonetheless, the effects of zirconia thicknesses of 0, 0.8, 1.0, and 1.2 mm should be further investigated. Additionally, in clinical situations, the degree of conversion of the marginal resin cement in crown restorations may surpass that of the resin cement irradiated through zirconia thicknesses exceeding 1.0 mm, as observed in this study. However, the marginal resin cement in the deep gingival sulcus might exhibit lower polymerization than the resin cement irradiated in vitro, owing to the risk of contamination by saliva and blood as well as inadequate light penetration. Therefore, this study, which demonstrates the biological effects of resin cement with reduced polymerization due to light attenuation through thick zirconia (such as 2.0 mm), can offer insights into understanding the inflammatory effects of resin cement with incomplete polymerization on periodontal tissue. Furthermore, the cells were cultured on the dual-cured resin cement discs in the culture medium in a 24-well plate. How this design mimics the human mouth in an in vivo situation is unknown. However, numerous studies cultured the cells on dental materials in a 24-well plate with the culture medium to evaluate the biological effects of dental materials in vitro [61,62,63,64]. Therefore, the results showing that the dual-cured resin cement with residual monomers induced inflammatory responses in the periodontal tissue cells are valuable for understanding the biological phenomenon around the margin of prostheses at a molecular level. On the other hand, the primary gingival fibroblasts from only one donor were used in this study because the primary cells from two donors, including those used in this study, showed similar inflammatory responses in the previous study [56]. However, the cells from multiple donors should be used in further studies. Moreover, one dual-cured resin cement and one zirconia disc were used in this study because one of the main purposes of this study was not to compare the effects of different resin cements on the cells but to reveal the mechanism by which resin cement induces inflammation in periodontal tissue using representative resin cement. We selected the SA Luting Multi containing Bis-GMA, TEGDMA, and HEMA as resin monomers because these monomers are contained in the most commercially available resin cement. For the same reason, we selected the Aadva Zirconia disc as the representative zirconia. However, the effects of multiple resin cements irradiated through multiple zirconia discs on cells should be compared in further studies.

Gingival fibroblasts play an important role as surveillants in periodontal tissue and as initiators of periodontitis [65,66]. Monocytes are innate immune cells that trigger inflammation by producing cytokines in periodontal tissue [67]. Resin cement is believed to induce periodontal inflammation through biofilm accumulation, stimulating innate immune cells [68]. On the other hand, the release of residual monomers from marginal cement or cement remnants through saliva and crevicular fluid might affect periodontal tissue. This is the first report to provide experimental evidence that dual-cured resin cement with incomplete polymerization directly induces proinflammatory cytokine release in hGFs and human monocytic cells. The study revealed that these reactions were induced by the differential time-dependent activation of MAP kinases. Our new findings may provide valuable clues for elucidating the pathological mechanism of periodontitis using dental resin materials and for fabricating novel adhesive materials in clinical settings.

## 4. Materials and Methods

### 4.1. Measurement of Total Light Transmittance of Zirconia Plate

Square zirconia plates (20 × 20 mm; Aadva Zirconia disc, ST, GC, Tokyo, Japan) were cut into thicknesses of 1.0, 1.5, and 2.0 mm. After sintering the zirconia plates according to the manufacturer’s instructions, the specimens were polished by grinding them on 320-, 400-, and 600-grit silicon carbide papers. They were then cleaned ultrasonically in distilled water for 5 min. The total light transmittance of the zirconia plates with different thicknesses was measured using a haze meter (NDH5000, Nippon Denshoku Industries, Tokyo, Japan).

### 4.2. Preparation of Dual-Cured Resin Cement Discs

A stainless mold with 13 mm inner diameter and 1 mm thickness on a glass plate covered with a polyester film was filled with dual-cured resin cement (PANAVIA SA Cement Universal, A2, Kuraray, Tokyo, Japan) following ISO 4049 standards [69] and a previous study [61] (Table 1). The filled resin cement was covered with a polyester film to prevent oxygen inhibition. It was irradiated for 10 s at five spots from above with or without zirconia plates (1.0 mm, 1.5 mm, and 2.0 mm) using an LED light-curing unit (1200 mW/cm^−2^, G-Light Prima Ⅱ plus, GC, Tokyo, Japan) (Figure 7a,b). For the no-irradiation group, the filled resin cement in the mold was incubated at 37 °C for 10 min. The dual-cured resin cement discs were polished by grinding them on a 1000-grit silicon carbide paper and cleaned ultrasonically in distilled water for 15 min.

### 4.3. Measurement of the DC of Dual-Cured Resin Cement

The DC of the prepared dual-cured resin cement discs was measured with Fourier transform infrared spectroscopy (FT/IR-6700, JASCO, Tokyo, Japan) with universal attenuated total reflectance (ATR). All ATR spectra were collected in the spectral range of 400–4000 cm^−1^ and 64 scans at 4 cm^−1^ resolution. DC was determined by comparing the ratio (R) between the reacted aliphatic bonds of the methacrylate functional group and the unreacted aromatic bonds of bisphenol. The spectra of unpolymerized cement (R_unpolym_) and polymerized cement (R_polym_) were detected by ATR. The aliphatic and aromatic bonds have characteristic infrared absorption peaks at 1637 and 1610 cm^−1^. The height of each peak was calculated using a correction baseline. The DC ratio was calculated using the following equation:DC (%) = 100 (1 − R_polym_/R_unpolym_)

### 4.4. Measurement of Vickers Hardness of Dual-Cured Resin Cement

The prepared dual-cured resin cement discs were placed on a microhardness tester (HM-102, Mitsutoyo, Kanagawa, Japan), and the diamond indenter was pressed into the surface of the discs at 0.20 kgf for 15 s. The indentation produced by the diamond was measured under a microscope, and the Vickers hardness was calculated using the following equation:HV = 1.8544F/d^2^
where HV is the Vickers hardness value, F is the test load (kgf), and d is the diagonal length of the indentation (mm).

### 4.5. Evaluation of Surface Topography of Dual-Cured Resin Cement

The surface roughness of the prepared dual-cured resin cement discs was evaluated using surface profilometry analysis. The Ra values of the specimens were measured using a non-contact profilometer (VR-5200; Keyence, Osaka, Japan).

The surface topography of the prepared dual-cured resin cement discs was observed using scanning electron microscopy (SEM, JSM-6390LA, JEOL, Tokyo, Japan). The specimens were sputter-coated with platinum, and SEM images were obtained at an acceleration voltage of 10 kV.

### 4.6. Measurement of Residual Resin Monomer Release of Dual-Cured Resin Cement

The prepared dual-cured resin cement discs were immersed in 5 mL ion-exchanged water at 37 °C for seven days. The peak of the resin monomers in the supernatant of the specimen immersed in water was detected by high-performance liquid chromatography (HPLC, Chromaster, Hitachi High-Tech, Tokyo, Japan) (Appendix A), and the concentrations of the released resin monomers in the water were calculated with standard solutions (10, 50, 100, and 250 ng/µL) of HEMA, Bisphenol A, TEGDMA, and Bis-GMA in acetonitrile.

### 4.7. Evaluation of Effects of Dual-Cured Resin Cement on Gene Expression of IL-1β, IL-6, PTGES2, TNFα, MMP2, and MMP9 in hGFs

The hGFs isolated from a healthy donor used in a previous study [70], which had been approved for human gingival tissue collection, subsequent genetic modification, genome/gene analyses, and the secondary use of cell sources by the Institutional Review Board at Osaka University Graduate School of Dentistry (approval number: H21-E7) and the Ethics Committee for Human Genome/Gene Analysis Research at Osaka University (approval number: 233), were used in this study. hGFs isolated from a healthy donor were expanded in DMEM with 15% FBS and 100 U penicillin/0.1 mg/mL streptomycin at 37 °C with 5% CO_2_ in a humidified incubator. The hGFs from passages 3 to 8 were used in this study. The hGFs were seeded on the prepared dual-cured resin cement discs in 24-well plates with 500 µL/well of the culture medium at a density of 1 × 10^5^ cells/well. Total RNA was extracted from cultured hGFs using the RNeasy Mini Kit (QIAGEN, Germantown, MD, USA) on day 2 and quantified using a Thermo Scientific NanoDrop 1000 ultraviolet-visible spectrophotometer (NanoDrop Technologies, Wilmington, DE, USA). After treatment with DNase I (Thermo Fisher Scientific, Waltham, MA, USA), cDNA was synthesized using the PrimerScript^TM^ 1^st^ Strand cDNA Synthesis Kit (Takara Bio, Shiga, Japan). mRNA expression was determined using the StepOnePlus Real-Time PCR system (Thermo Fisher Scientific) and Thunderbird SYBR qPCR Mix (Toyobo, Osaka, Japan) or the TaqMan^TM^ Gene Expression Master Mix (Thermo Fisher Scientific) for SYBR-green-based PCR or TaqMan probe-based PCR, respectively. Target gene expression was quantitatively analyzed using the ΔΔCT method. *GAPDH* was used as the housekeeping gene. The primers used are listed in Appendix A.

### 4.8. Evaluation of Effects of Dual-Cured Resin Cement on Gene Expression of TNFα, NOS2, IL-10, and ARG1 in THP-1 Cells

THP-1 cells (JCRB0112.1; Japanese Collection of Research Bioresources Cell Bank, Osaka, Japan), the human monocytic cell line, were expanded in DMEM with 10% FBS and 100 U penicillin/0.1 mg/mL streptomycin at 37 °C and 5% CO_2_ in a humidified incubator. THP-1 cells were seeded onto the prepared dual-cured resin cement discs in 24-well plates at a density of 1 × 10^5^ cells/well. Total RNA was extracted from cultured THP-1 cells on day 2, and mRNA expression was evaluated using SYBR-green-based PCR as described above. The primers used are listed in Appendix A.

### 4.9. Cell Proliferation Assay

The cells were seeded on the prepared dual-cured resin cement discs in 24-well plates at a density of 1 × 10^5^ cells/well. On days 1, 2, and 3, the culture medium was changed to 10% WST-1 reagent (Roche, Basel, Switzerland) and incubated for 1.5 h. Absorbance was measured at 405 nm using an iMark Microplate Absorbance Reader (Bio-Rad Laboratories, Inc., Hercules, CA, USA).

### 4.10. DNA Quantification

One hour after the cells were seeded on the prepared dual-cured resin cement discs in 24-well plates, they were washed with PBS. The total DNA in the culture was measured using a DNA quantification kit (COSMO BIO Co., Ltd., Tokyo, Japan) according to the manufacturer’s protocol. After the cell lysis buffer was added to each well, the cell lysate was mixed with cell lysis buffer and Hoechst 33258. The mixture was transferred to 96-well black microplates, and the fluorescence intensity was measured using a GloMax-Multi Detection System reader (Promega Corporation, Madison, WI, USA) (excitation: 365 nm; emission: 410–460 nm).

### 4.11. Evaluation of Intracellular ROS Level

One hour after the cells were seeded on the prepared dual-cured resin cement discs in 24-well plates, they were washed with PBS. The cells were incubated in PBS containing 10 µM 2′,7′-dichlorodihydrofluorescein diacetate (H2DCFDA) (Thermo Fisher Scientific) for 15 min at 37 °C. The fluorescence intensity of dichlorofluorescein diacetate (DCFDA) was measured using a SpectraMax M2/M2e microplate reader (Molecular Devices, San Jose, CA, USA) (excitation: 490 nm; emission: 510 nm). Intracellular ROS levels were determined according to fluorescence intensity values normalized to DNA concentrations.

### 4.12. Western Blotting Analysis of MAP Kinases

On days 1, 2, and 3, one hour after the cells were seeded on the prepared dual-cured resin cement discs in 24-well plates, the cells were washed with PBS. The cells were lysed in EBC buffer (50 mM Tris 7.5, 120 mM NaCl, 0.5% NP-40) supplemented with a protease inhibitor cocktail (Sigma-Aldrich, St. Louis, MO, USA) and phosphatase inhibitor tablets (Sigma-Aldrich). The protein concentration of the lysates was measured using the Bio-Rad protein assay reagent (Bio-Rad Laboratories, Inc., Hercules, CA, USA). Identical amounts of whole-cell lysates were resolved by 10% SDS-polyacrylamide gel electrophoresis and transferred to a polyvinylidene difluoride membrane (Immun-Blot PVDF Membrane, Bio-Rad Laboratories, Inc.). The blots were blocked with 5% skim milk (Becton Dickinson, Franklin Lakes, NJ, USA) and then incubated with primary anti-phospho-p44/42 MAPK (Erk1/2) antibody (1:2000; Cell Signaling, Danvers, MA, USA), anti-phospho-p38 MAPK antibody (1:2000; Cell Signaling), anti-phospho-SAPK/JNK antibody (1:1000; Cell Signaling), anti-p44/42 MAPK (Erk1/2) antibody (1:2000; Cell Signaling), anti-p38 MAPK antibody (1:1000; Cell Signaling), anti-JNK antibody (1:100; Santa Cruz Biotechnology, Dallas, TX, USA), or anti-β-actin polyclonal antibody (1:5000; Cell Signaling Technology) at 4 °C overnight. After washing with TBST (10 mM Tris-HCl pH 7.4, 100 mM NaCl, and 0.1% Tween), the membranes were incubated with horseradish-peroxidase-conjugated secondary antibodies (Santa Cruz Biotechnology) for 1 h at room temperature. Finally, the signals were visualized using horseradish peroxidase substrate (Merck Millipore, Burlington, MA, USA).

### 4.13. Statistics and Reproducibility

One-way analysis of variance with Tukey’s multiple comparison test was performed for statistical analyses. A statistically significant difference was defined as *p* < 0.05. The sample size and the number of replicates are reported in the figure legends. The data are shown as mean and standard deviation.

## Figures and Tables

**Figure 1 ijms-24-09861-f001:**
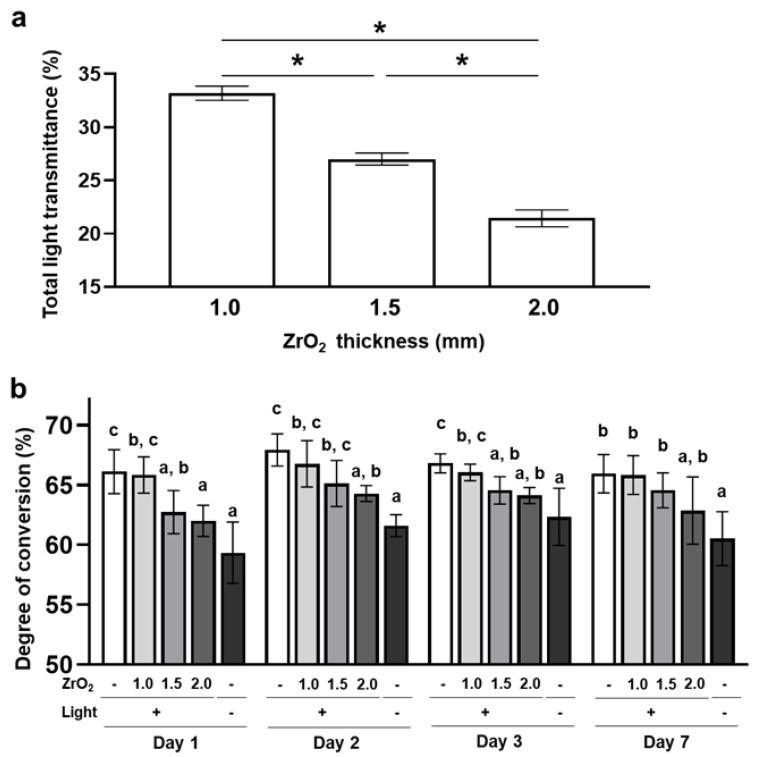
Effects of zirconia thickness on the degree of conversion of the dual-cured resin cement through light attenuation: (**a**) The total light transmittance of zirconia plates (ZrO_2_) with different thicknesses was measured with a haze meter (*n* = 10). (**b**) The degree of conversion of dual-cured resin cement with irradiation through different thicknesses of zirconia or without irradiation was determined by Fourier transform infrared spectroscopy with universal attenuated total reflectance (*n* = 7). ANOVA with Tukey’s multiple-comparison test on the same day (**a**,**b**). Data are presented as mean values ± SD; *p* < 0.05 was considered significant. Asterisks and different letters indicate statistically significant differences between multiple groups.

**Figure 2 ijms-24-09861-f002:**
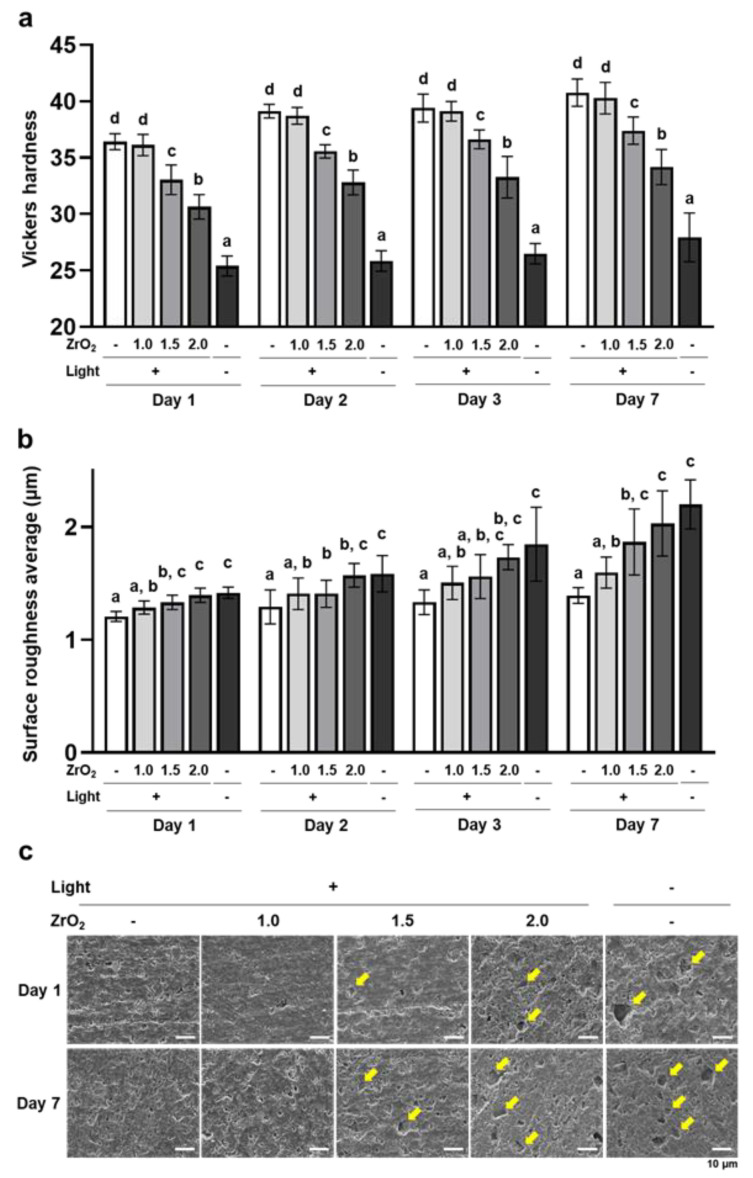
Effects of zirconia thickness on mechanical and physical properties of the dual-cured resin cement through light attenuation: (**a**) The Vickers hardness of dual-cured resin cement prepared in each condition was measured with a microhardness tester (*n* = 10). (**b**) The surface roughness average of dual-cured resin cement was determined using an atomic force microscope (*n* = 7). (**c**) Representative secondary scanning electron microscope images of dual-cured resin cement on days 1 and 7 (scale bars: 10 µm). Yellow arrows indicate microcavities. ANOVA with Tukey’s multiple-comparison test on the same day (**a**,**b**). Data are presented as mean values ± SD; *p* < 0.05 was considered significant. Different letters indicate statistically significant differences between multiple groups.

**Figure 3 ijms-24-09861-f003:**
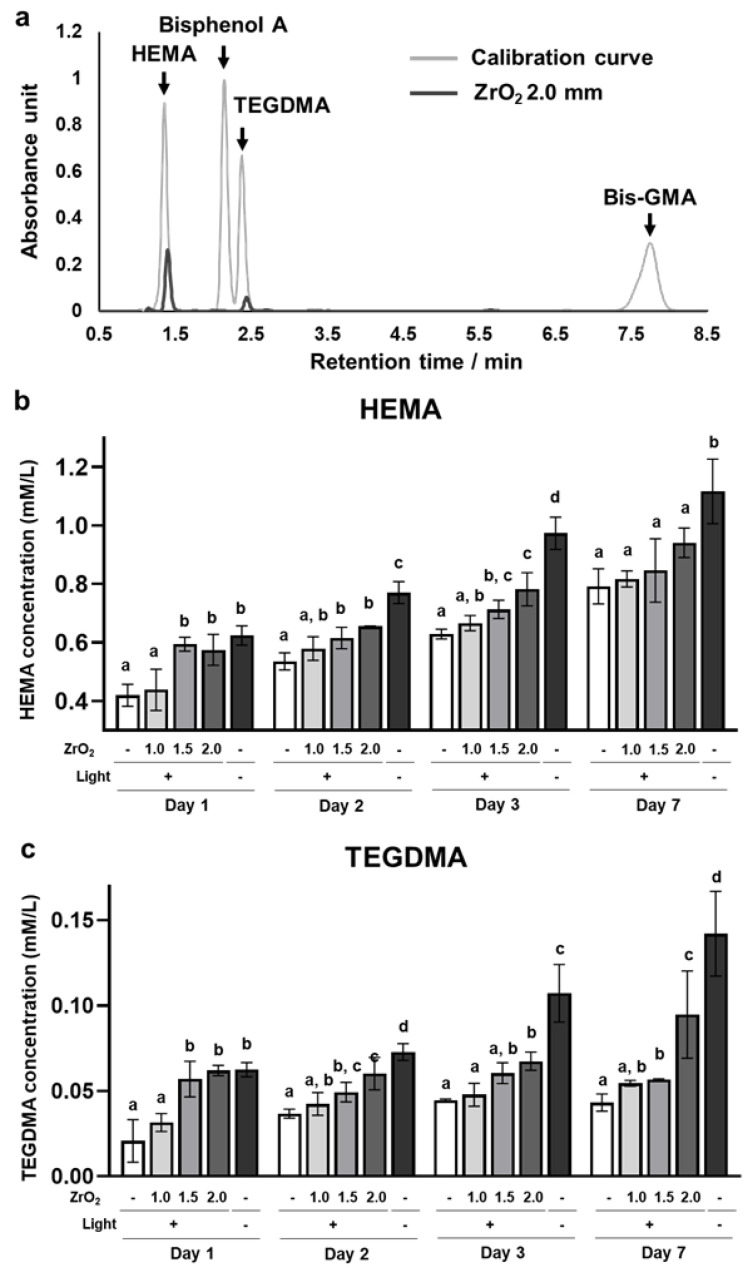
Effects of zirconia thickness on resin monomer release of the dual-cured resin cement through light attenuation: (**a**) The representative chromatograms of HEMA, Bisphenol A, TEGDMA, and Bis-GMA standard solutions and the supernatant of the dual-cured resin cement irradiated through 2.0 mm thickness of zirconia immersed in water on day 1. The concentration of released (**b**) HEMA and (**c**) TEGDMA in the supernatant of the dual-cured resin immersed in the water (*n* = 7). ANOVA with Tukey’s multiple-comparison test on the same day (**b**,**c**). Data are presented as mean values ± SD; *p* < 0.05 was considered significant. Different letters indicate statistically significant differences between multiple groups.

**Figure 4 ijms-24-09861-f004:**
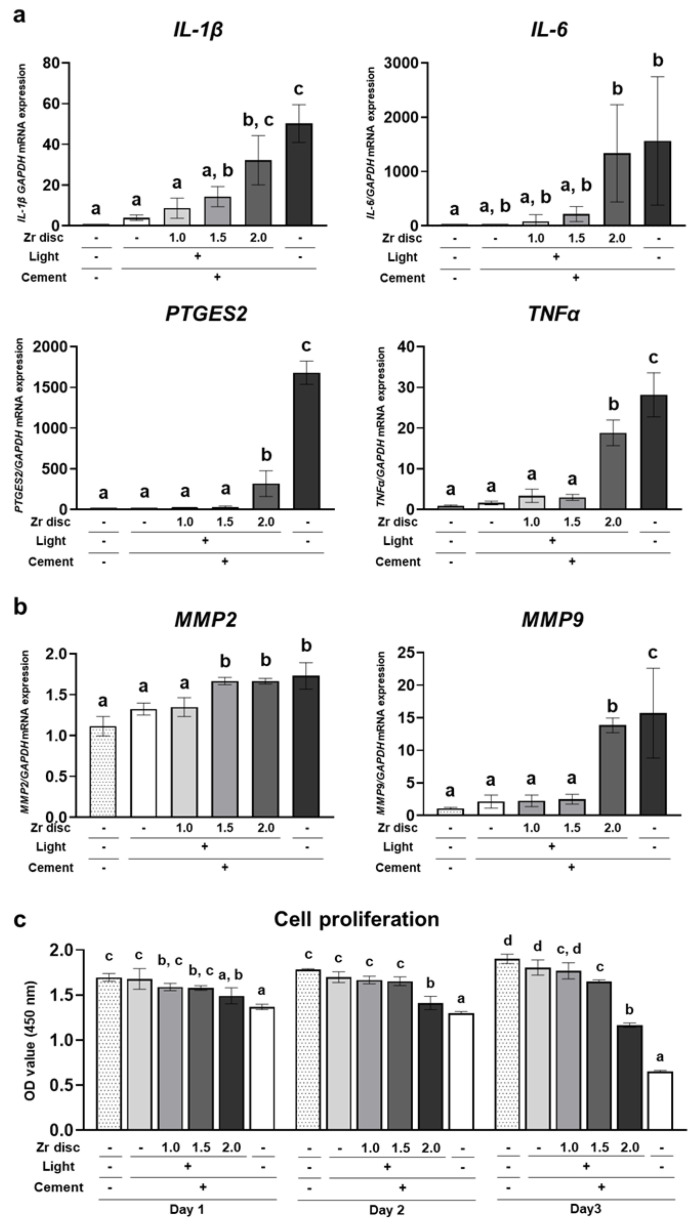
Effects of the dual-cured resin cement with incomplete polymerization on human gingival fibroblasts (hGFs). Gene expression of (**a**) proinflammatory cytokines *IL-1β*, *IL-6*, *PTGES2,* and *TNFα* and (**b**) matrix metalloproteinases *MMP2* and *MMP9* relative to *GAPDH* in hGFs cultured on the dual-cured resin cement on day 2 was determined by quantitative real-time RT-PCR analysis (*n* = 3). (**c**) WST-1-based evaluation of cell proliferation of hGFs cultured on the dual-cured resin cement (*n* = 3). ANOVA with Tukey’s multiple-comparison test on the same day. Data are presented as mean values ± SD; *p* < 0.05 was considered significant. Different letters indicate statistically significant differences between multiple groups.

**Figure 5 ijms-24-09861-f005:**
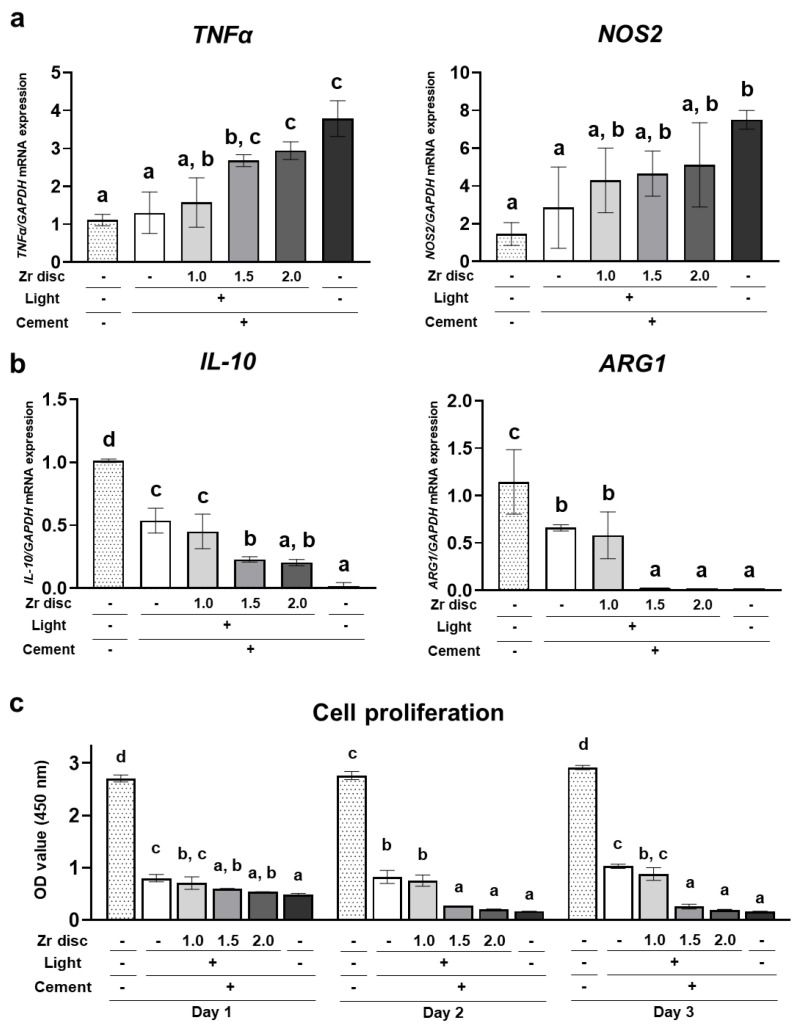
Effects of the dual-cured resin cement with incomplete polymerization on THP-1 cells. Gene expression of (**a**) proinflammatory cytokines *TNFα* and *NOS2* and (**b**) anti-inflammatory cytokines *ARG1* and *IL-10* relative to *GAPDH* in THP-1 cells cultured on the dual-cured resin cement on day 1 was determined by quantitative real-time RT-PCR analysis (*n* = 3). (**c**) WST-1-based evaluation of cell proliferation of THP-1 cells cultured on the dual-cured resin cement (*n* = 3). ANOVA with Tukey’s multiple-comparison test on the same day. Data are presented as mean values ± SD; *p* < 0.05 was considered significant. Different letters indicate statistically significant differences between multiple groups.

**Figure 6 ijms-24-09861-f006:**
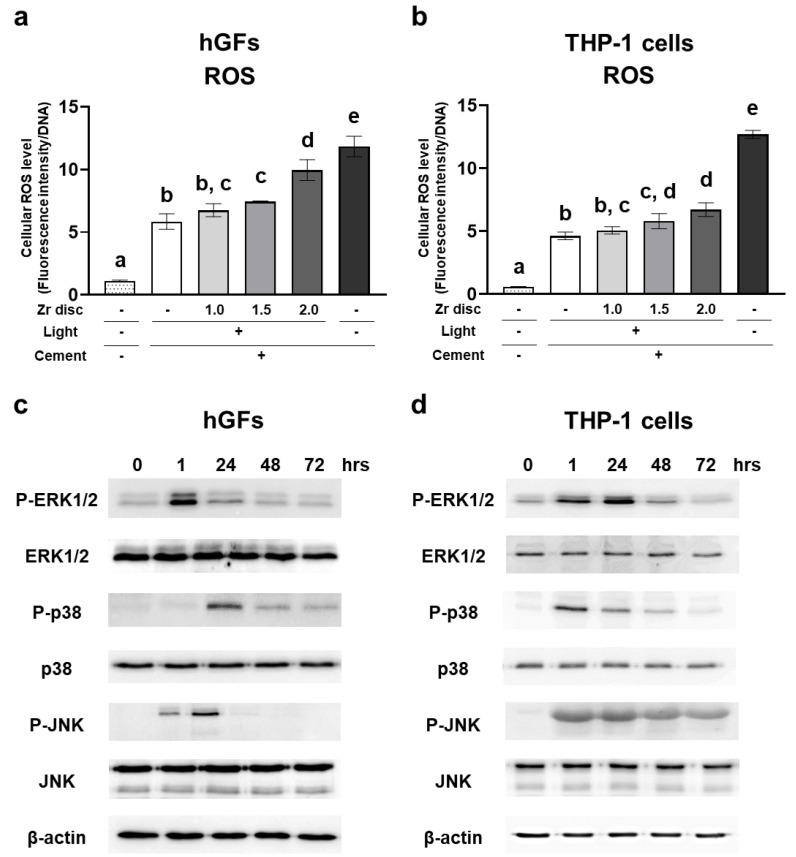
Effects of the dual-cured resin cement with incomplete polymerization on intracellular reactive oxygen species (ROS) generation and mitogen-activated protein (MAP) kinase activation of hGFs and THP-1 cells. Intracellular ROS levels of (**a**) hGFs and (**b**) THP-1 cells cultured on the dual-cured resin cement were measured and normalized to the DNA concentrations at 1 h (*n* = 3). Activation of MAPK kinases of (**c**) hGFs and (**d**) THP-1 cells cultured on the dual-cured resin cement. ANOVA with Tukey’s multiple-comparison test at the same time (**a**,**b**). Data are presented as mean values ± SD; *p* < 0.05 was considered significant. Different letters indicate statistically significant differences between multiple groups.

**Figure 7 ijms-24-09861-f007:**
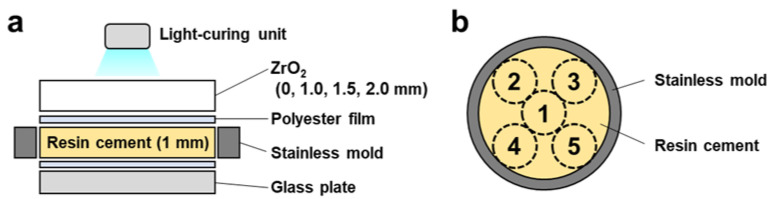
Scheme of the dual-cured resin cement preparation: (**a**) The dual-cured resin cement was filled in the stainless mold on the glass plate. The filled resin cement was covered with a polyester film and irradiated with or without zirconia plates using an LED light-curing unit. (**b**) The cement was irradiated for 10 s at five spots from above.

**Table 1 ijms-24-09861-t001:** Composition of the dual-cured resin cement.

Material	Composition
PANAVIA SA Cement Universal, A2, Kuraray, Tokyo, Japan	Bis-GMA, TEGDMA, HEMA, sodium fluoride, silanated barium glass filler, silanated colloidal silica, aluminum oxide filler, 10-MDP, hydrophobic aromatic dimethacrylate, silane coupling agent, camphorquinone, peroxide, accelerators, catalysts, pigments

## Data Availability

The data presented in this study are available on request from the author.

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
