# Peer review of "Incomplete Polymerization of Dual-Cured Resin Cement Due to Attenuated Light through Zirconia Induces Inflammatory Responses"

_ijms, 2023, doi:10.3390/ijms24129861_

Round 1

Reviewer 1 Report

It is an innovative and well-designed study, following revisions are required before publication:

1. please add reference support to the method of "4.2. Preparation of dual-cured resin cement discs"

2. please indicate the passage number of hGFs used in this study, since they are primary cells. 

3. for in vitro cell studies, the ratio of medium/disk is important, please explain the rational of choosing 24-well plates for cell culture and indicate how much cell culture medium was added into each well, how this design mimic the human mouth in vivo situation. 

4. it makes sense using primary cells in this study, the only limitation is the donor number, cells were all from one donor

Reviewer 2 Report

Materials and Methods should be before Results and Discussion. The results can't be followed. 

The Y-TZP can be from 3-5 wt% Y2O3 and multilayered today . This one is more suitable for layering (reduced thickness and ceramic layering).  You are mentioning implants but implants are cemented with different cements and cellular respond is not the same. 

You don't need resine cement to be better after resin cementation. It could be cemented conventionally. This statement is for glass-ceramics. 

The biggest issue of the paper is why are you compering thickness of zirconium oxide and periodontal cells? Periodontal cells means that cement is out of the restoration and then tickness don't meter.

The best possible degree of conversion is on the margin of the restoration where the thickness is the lowest. The influence of the soft tissue maybe have differece to the degree of converison in periodontal tissue.

The only reason to do this research is maybe influence to pulp cells if preparation is very deep.

As I mentioned, you have different Y-TZP on the market today and dentistry is using more and more monolitic solutions digitaly done. That means reduction of the thickness of the preparations and restorations. Above 1.5 mm are crowns mostly on implants and there is no pulp and the soft tissue block the light if you are subgingival not restoration.

Table from Methods is in discussion 284

Why just this cement? For me it would be better to compere different cements. All results are based on 1 zirconium oxide and 1 cement.

In discussion you are saying: Y-TZP zirconia, even with a 0.5 mm thickness, exhibited a high 253 fracture resistance (1,421 N) [31]. On the other hand you are mentioned resin cementation for better mechanical properties...

Please check numbers in the results regarding thicknesses.

Round 2

Reviewer 2 Report

The paper is improved by explaining the limitations more in details. The implants are excluded.

However, the marginal thickness of the crown should not be more than 1 mm. The point of the research should be 0, 0.8, 1.0, maybe 1.2. Respect to other papers, but there is no purpose in 2 mm for cell response. For mechanical properties and conversion this is maybe good approach. 

Results are 3.0 not 2.0.

The suggested papers about marginal adaptations are too old. The precision of zirconia crowns are much better now. 

This part of the cement and margin probably would  be  properly cured.
